# Vulnerability Assessment of Six Endemic Tibetan-Himalayan Plants Under Climate Change and Human Activities

**DOI:** 10.3390/plants14152424

**Published:** 2025-08-05

**Authors:** Jin-Dong Wei, Wen-Ting Wang

**Affiliations:** School of Mathematics and Computer Science, Northwest Minzu University, Lanzhou 730030, China; y231530324@stu.xbmu.edu.cn

**Keywords:** climate change, Climate Niche Factor Analysis, human activities, protected areas, vulnerability

## Abstract

The Tibetan-Himalayan region, recognized as a global biodiversity hotspot, is increasingly threatened by the dual pressures of climate change and human activities. Understanding the vulnerability of plant species to these forces is crucial for effective ecological conservation in this region. This study employed an improved Climate Niche Factor Analysis (CNFA) framework to assess the vulnerability of six representative alpine endemic herbaceous plants in this ecologically sensitive region under future climate changes. Our results show distinct spatial vulnerability patterns for the six species, with higher vulnerability in the western regions of the Tibetan-Himalayan region and lower vulnerability in the eastern areas. Particularly under high-emission scenarios (SSP5-8.5), climate change is projected to substantially intensify threats to these plant species, reinforcing the imperative for targeted conservation strategies. Additionally, we found that the current coverage of protected areas (PAs) within the species’ habitats was severely insufficient, with less than 25% coverage overall, and it was even lower (<7%) in highly vulnerable regions. Human activity hotspots, such as the regions around Lhasa and Chengdu, further exacerbate species vulnerability. Notably, some species currently classified as least concern (e.g., *Stipa purpurea* (*S. purpurea*)) according to the IUCN Red List exhibit higher vulnerability than species listed as near threatened (e.g., *Cyananthus microphyllus* (*C. microphylla*)) under future climate change. These findings suggest that existing biodiversity assessments, such as the IUCN Red List, may not adequately account for future climate risks, highlighting the importance of incorporating climate change projections into conservation planning. Our study calls for expanding and optimizing PAs, improving management, and enhancing climate resilience to mitigate biodiversity loss in the face of climate change and human pressures.

## 1. Introduction

As the highest and largest plateau ecosystem globally, the Tibetan-Himalayan region is a critical biodiversity hotspot characterized by unique topographic and climatic features [1,2,3,4]. This region encompasses two biodiversity hotspots, the Heng Duan Mountains (HDMs) and the Himalayas, which are characterized by high species endemism and ecological sensitivity [5,6,7]. The Tibetan-Himalayan region harbors numerous endemic and endangered species, many of which inhabit alpine environments above 3000 m and have adapted to extreme climates and specialized ecological niches [8,9,10]. The survival of these species not only reflects the ecological integrity of local ecosystems but also reveals the overall ecological vulnerability of the plateau [11,12]. Climate change has emerged as a major driver of global biodiversity loss and ecosystem degradation [13,14,15], and it is projected to surpass other threats, such as habitat destruction, in its impact on biodiversity in the future [15,16]. Alpine species, which are tightly connected to low-temperature environments, are particularly sensitive to climate warming [17,18,19]. In the Tibetan-Himalayan region, one of the world’s most climate-sensitive regions, surface air temperatures are projected to increase at approximately 1.5 times the global average rate [20,21]. This accelerated warming drives a progressive reduction in the extent of suitable habitat for alpine species, which may exacerbate habitat fragmentation [13,22]. Consequently, many endangered species that rely on specific environmental conditions are facing increasing survival pressures, with some populations even confronting the risk of local extinction [22,23].

In addition to climate change, human activities impose profound pressures on biodiversity [24,25]. Among these, land use change, particularly habitat conversion and degradation, has emerged as the primary driver of global biodiversity loss [25,26,27]. Recent studies estimate that such processes have reduced global average species richness by an average of 13.6% and the total biological abundance by up to 10.7% [25]. Habitat fragmentation further exacerbates these effects, leading to localized biodiversity declines ranging from 13% to 75%, and disrupting vital ecosystem functions such as nutrient cycling [28]. Alpine ecosystems are particularly vulnerable, with endemic plants like *Rhodiola* and *Meconopsis* experiencing dramatic population declines due to overharvesting [29,30]. Moreover, alpine areas face the twin threats of climate change and environmental pollution [16,31,32], including the deposition of organic pollutants and the eutrophication of water bodies [33]. Collectively, these pressures have forced many alpine species to migrate to higher altitudes or latitudes, with some species even facing the risk of local extinction [13,30]. Therefore, investigating the extinction risks of endemic species on the Tibetan-Himalayan region under climate change and human activity impacts will not only help understand the response mechanisms of alpine ecosystems but also provide a scientific basis for prioritizing regional biodiversity conservation.

Currently, global assessments of endangered species rely primarily on the IUCN Red List Criteria established by the International Union for the Conservation of Nature, which serves as the most authoritative international standard for evaluating biodiversity conservation status [34,35,36]. China has also developed a national red list for its domestic species using the same criteria. However, the IUCN Red List assessment criteria primarily focus on species population size and human-induced habitat pressures, without adequately accounting for potential future threats such as climate change [14,29,37,38]. To address this limitation, Climate Change Vulnerability Assessment (CCVA) frameworks offer a more comprehensive approach to evaluating species vulnerability to climate change. CCVA integrates three key components: sensitivity (intrinsic traits that influence species’ response to climate change, such as ecological niche specialization and environmental tolerance), adaptive capacity (the ability to cope with environment change through mechanisms like phenotypic plasticity, dispersal, or genetic diversity), and exposure (the extent of climate change within species’ habitats) [14,39]. Recently, Rinnan and Lawler proposed Climate Niche Factor Analysis (CNFA), a comprehensive framework that uses species distribution and bioclimatic data to separate species sensitivity from habitat exposure, revealing spatial patterns of climate vulnerability [40]. CNFA has been widely applied to assess the vulnerability of plants [29,30,41,42] and mammals [15,43] to climate change. However, some studies have noted that in CNFA, sensitivity indicators often play a more decisive role than exposure indicators in assessing species vulnerability [15,43]. This imbalance likely arises from differences in scale between sensitivity and exposure indicators within CNFA, as sensitivity metrics typically have a broader range than exposure indicators, leading to vulnerability assessments being disproportionately influenced by sensitivity.

In addition, the establishment of protected areas (PAs) is a crucial strategy for biodiversity conservation, playing an essential role in safeguarding species survival, enhancing ecosystem functions, and maintaining critical ecological services [44,45,46,47]. Previous research has shown that PAs not only facilitate ecological restoration but also contribute to improving the livelihoods of neighboring communities [48,49,50]. However, the global network of PAs faces persistent challenges, including insufficient coverage, uneven spatial distribution, and inadequate management [44,47,51,52,53]. Many endangered species and ecologically valuable regions remain under protection, while some existing PAs fail to achieve their conservation objectives due to insufficient long-term investment and ineffective management [44,54,55,56]. It is therefore critical to account for the uneven distribution and limitations of PAs when conducting species vulnerability assessments.

In this study, we investigate the vulnerability of species in the Tibetan-Himalayan region to the combined impacts of climate change and human activities by focusing on six representative alpine endemic herbaceous species. To address scale discrepancies between sensitivity and exposure metrics within the CNFA framework, we applied max-min normalization of the sensitivity indicators across all species, as well as the exposure indicators, under various carbon emission scenarios. Using this improved CNFA approach, we quantified the climate change vulnerability of the six species and analyzed their spatial vulnerability patterns. Furthermore, by integrating human footprint data with the distribution of existing PAs, we identified the influence of human activities, delineated priority conservation zones, and highlighted gaps in current protection efforts. Through this comprehensive assessment of species vulnerability, our study provides a scientific basis for developing targeted conservation strategies aimed at mitigating the impacts of climate change and human activity on alpine endemic species.

## 2. Results

### 2.1. Climate Change Vulnerability of Different Species

The results from the CNFA indicate that six plant species from the Tibetan-Himalayan region showed signs of sensitivity, exposure, and vulnerability among each bioclimatic variable. Specifically, all six species showed high sensitivity to precipitation of the warmest quarter (PWQ) and low sensitivity to isothermality (ISO) (Table 1). In addition, *Androsace erecta Maxim.*, *Incarvillea younghusbandii Sprague*, and *Anaphalis xylorhiza Sch.-Bip. ex Hook. f.* showed high sensitivity to the mean temperature of the warmest quarter (MTWQ), while *Syncalathium kawaguchii (Kitam.) Y. Ling*, *Cyananthus microphyllus Edgew.*, and *Stipa purpurea Griseb.* showed high sensitivity to the mean diurnal range (MDR), precipitation seasonality (PS), and precipitation of the coldest quarter (PCQ), respectively (Table 1). Among the six plant species, *C. microphyllus* had the greatest total marginality, and *A. erecta* had the least total marginality. However, *S. kawaguchii* had the greatest total sensitivity, and *A. xylorhiza* had the least total sensitivity (Table 2).

The six plant species consistently showed high levels of exposure to the MDR and MTWQ and low levels of exposure to the PCQ and PS. In addition, the exposures to each bioclimatic variable for the habitat of the six plant species from present day to 2050 were generally larger under high carbon emission scenarios than under low carbon emission scenarios (Table 1). The total exposure was greatest for *I. younghusbandii* and the least for *A. erecta* among the six plant species. In addition, the total exposure of all species was higher in the high carbon emission scenario than in the moderate carbon emission scenario (Table 2).

Similar to the sensitivity indicators, all six plant species showed significant vulnerability to the PWQ across their ranges. Four of these species—*A. erecta*, *S. purpurea*, *I. younghusbandii* and *A. xylorhiza*—additionally showed significant vulnerability to the MTWQ. On the other hand, *S. kawaguchii* and *C. microphyllus* showed significant vulnerability to the MDR and PS. Notably, all six plant species exhibited low vulnerability to the PCQ. Furthermore, in the projected future climate change scenarios up to 2050, the vulnerability factors of each bioclimatic variable were generally higher under the high carbon emission scenario than the corresponding values under the moderate carbon emission scenario (Table 1). For the total vulnerability of each species, we found, after normalization, that *S. kawaguchii* had the greatest overall vulnerability and *A. xylorhiza* had the least. In addition, the total vulnerability of all species was higher in the high carbon emission scenario than in the moderate carbon emission scenario. Among them, the total vulnerability of *C. microphyllus* was lower than that of *S. purpurea* and *I. younghusbandii*, but the endangerment rating of *C. microphyllus* was near threatened (NT), while the endangerment rating of *S. purpurea* and *I. younghusbandii* was least concern (LC). This suggests that conservation assessments of *C. microphyllus* in the context of future climate change need to pay particular attention to the potential threats posed by climate change, and that it is necessary to incorporate the impacts of future climate change on the species into the assessment metrics of the red list (Table 1 and Table 2; Figure 1) [30,40,42].

### 2.2. Spatial Patterns of Sensitivity, Exposure, and Vulnerability to Climate Change

Overall, the spatial sensitivity distribution maps showed that the spatial sensitivities of these six species showed a gradient trend of being high in the south and low in the north, and the sensitivities were generally low in most areas of their habitats (dominated by green in Figure 2). Specifically, *S. kawaguchii* and *A. erecta* showed high sensitivity in most areas of their habitats compared with the other four species, especially around the Himalayas and in the central Tibetan-Himalayan region. In addition, *A. erecta* also showed high sensitivity in the vicinity of the HDMs. *C. microphyllus*, *S. purpurea*, *I. younghusbandii*, and *A. xylorhiza* showed high sensitivity in the southern Himalayas. Meanwhile, *C. microphyllus* and *I. younghusbandii* also showed high sensitivity in the central Tibetan-Himalayan region, while *S. purpurea* showed high sensitivity near the HDMs. Compared with the spatial sensitivity of *S. kawaguchii* and *A. erecta*, *C. microphyllus*, *S. purpurea*, *I. younghusbandii*, and *A. xylorhiza* showed lower sensitivity in most areas of their habitats (Figure 2).

Under future climate changes, the spatial exposure of these six species tended to be high in the north and low in the south within their habitats, with low spatial exposure, especially near the Himalayas and the east-central Tibetan-Himalayan region. *C. microphyllus*, *S. purpurea*, *I. younghusbandii*, and *A. xylorhiza* showed higher spatial exposure in most parts of their habitats compared with *S. kawaguchii* and *A. erecta* (Figure 3). In addition, the spatial exposure of these six species in the same areas was higher in the SSP5-8.5 scenario compared with the SSP2-4.5 scenario (Appendix A).

From the spatial vulnerability distribution map, it can be seen that the spatial vulnerability of these six species showed a distribution pattern of being high in the west and low in the east. In addition, in the central Tibetan-Himalayan region and the HDMs, all six species showed low vulnerability to climate change. Specifically, *S. kawaguchii* and *A. erecta* showed significant vulnerability in most areas of their habitats (red and light green colors dominated their habitats in Figure 4). In contrast, the vulnerability of *C. microphyllus*, *S. purpurea*, *I. younghusbandii*, and *A. xylorhiza* was higher in the western part of the Tibetan-Himalayan region and gradually decreased in the eastern direction. *C. microphyllus*, *S. purpurea*, and *I. younghusbandii* showed extremely low vulnerability to climate change in the eastern part of their habitats, while *A. xylorhiza* showed extremely low vulnerability to climate change throughout its habitat. Overall, *S. kawaguchii* had the highest spatial vulnerability, while *A. xylorhiza* had the lowest spatial vulnerability (Figure 4). For the different SSPs, the areas of high vulnerability in the SSP2-4.5 scenario also showed higher vulnerability in the SSP5-8.5 scenario (Appendix A).

### 2.3. Integrated Vulnerability of Coupled Human Footprints and Protected Area Distribution

The vulnerability of these six species within their habitat ranges showed a distribution pattern of being high in the west and low in the east. In addition, the climate change vulnerability of these six species was generally low in the central Tibetan-Himalayan region and the HDMs. Based on data on the percentage of area covered by protected areas in relation to species habitat, the protected area coverage for these six species in the Tibetan-Himalayan region was generally low, with none exceeding 25%. Among them, *S. kawaguchii* and *A. erecta*, which are assessed as NT species by the Red List of Chinese Plants database, had extremely low habitat coverage in nature reserves, being 8.55% and 12.48%, respectively, and in the areas with the highest vulnerability level, the coverage of their protected areas was only 3.25% and 0.67%, respectively. However, *C. microphyllus*, which was assessed as NT and had the smallest habitat area, had the highest habitat coverage of the six species in nature reserves at 25.02%; however, in the area with the highest vulnerability rating, it had only 5.72% protected area coverage. Overall, the habitats of the three species assessed as NT generally had low coverage in nature reserves, and none of the areas with the highest vulnerability ratings had more than 6% protected area coverage (Table 3; Figure 5; Appendix A).

For *S. purpurea*, *I. younghusbandii*, and *A. xylorhiza*, which were assessed as LC by the Red List of Chinese Plants database, *S. purpurea* and *I. younghusbandii* accounted for a relatively high proportion of the areas with high vulnerability ratings, amounting to 25.3% and 43.6%, respectively. However, their habitat coverage in nature reserves was low, being 15.52% and 16.29%, respectively, and in the areas with the highest vulnerability rank, their coverage in protected areas was only 5.02% and 6.95%, respectively. The percentage of *A. xylorhiza* in the areas with high vulnerability was extremely low at only 0.4%; its habitat coverage in nature reserves was 19.23%, which is moderate; and in the areas with the highest vulnerability, its protected area coverage was as low as 0.01% (Table 3; Figure 5; Appendix A). For the NT species *C. microphyllus*, *S. purpurea* and *I. younghusbandii* showed significant vulnerability, with *I. younghusbandii* having a significantly higher proportion of its distribution in areas of higher vulnerability levels than *C. microphyllus*. It is worth noting that in areas of the highest vulnerability levels, *I. younghusbandii* also had slightly higher coverage of protected areas than *C. microphyllus*. Notably, *I. younghusbandii* also had a slightly higher percentage of protected area coverage than *C. microphyllus* in the areas with the highest vulnerability ratings. This indicates that, under future climate change scenarios, the conservation assessment of endangered plants should not only focus on the potential threats posed by climate change but also prioritize enhancing the coverage of natural protected areas [44,47].

As can be seen from the coupled human footprint and climate change species vulnerability maps, the habitats of these six species were affected by human footprints to varying degrees in different regions, especially in the regions around densely populated metropolitan areas such as Lanzhou, Chengdu, Lhasa, and New Delhi, where the impacts of human activities have been particularly significant, leading to a further increase in the vulnerability of the species in these regions (Figure 6). The impact of the human footprint on species vulnerability was more pronounced in the high carbon emission scenario (Appendix A).

## 3. Discussion

Accurate and scientifically robust predictions of which species are most vulnerable to climate change and which regions face the greatest risks are critical for informing effective conservation strategies, especially for endangered and threatened species [43,57,58,59]. Our results indicated significant differences among the six species in terms of sensitivity, exposure, and vulnerability. Sensitivity showed a clear gradient, being high in the southern areas and low in the northern areas. Exposure displayed the opposite pattern, being high in the north and low in the south. Meanwhile, vulnerability was predominantly high in the western regions and low toward the east. At the species level, *S. kawaguchii* displayed the highest sensitivity and vulnerability, whereas *A. xylorhiza* showed the lowest levels in both metrics. In terms of exposure, *I. younghusbandii* was the most exposed species, while *A. erecta* was the least exposed. These spatial patterns of species vulnerability are primarily driven by the combined spatial variation in sensitivity and exposure [40].

Although *S. kawaguchii* and *A. xylorhiza* ranked highest and lowest in sensitivity and vulnerability, respectively, the spatial distribution of vulnerability within their habitats did not fully align with that of sensitivity. This discrepancy can be attributed to the normalization applied to sensitivity and exposure when calculating vulnerability. Normalization is designed to prevent any single factor from disproportionately influencing the outcome, thereby enabling a more balanced and integrated reflection of both components.

Across all six species, the spatial distribution of climate change vulnerability consistently showed higher values in the western portions of their habitats and lower values in the east. Notably, the central Tibetan-Himalayan region and the HDMs exhibited relatively low levels of climate change vulnerability across these species, suggesting their potential role as future climate refuge locations. This observation aligns with the findings of Wang et al. (2021) [30], reinforcing the critical ecological function of these regions in conserving alpine biodiversity under future climate change.

For areas identified as critically vulnerable to climate change, we recommended the implementation of assisted migration strategies for the six species, relocating them to more climatically suitable habitats in the future [60,61]. Semi-natural habitats, such as botanical gardens, which provide appropriate growing conditions, present promising relocation sites [62,63,64]. Another viable approach is to establish nature reserves in these climate-vulnerable zones. However, given the current limitations of the PA system, such as insufficient coverage, spatial imbalance, and underfunded management, implementing new reserves in highly vulnerable regions remains challenging and demands long-term planning and multi-stakeholder collaboration [44,54,55].

Furthermore, current evaluations based on the IUCN Red list Criteria classify *C. microphyllus* as a near-threatened (NT) species, indicating a significantly higher risk compared with *S. purpurea* and *I. younghusbandii*, which are classified as least concern (LC) species. However, our climate change vulnerability assessment results indicate that under future climate change, *C. microphyllus* actually exhibits lower vulnerability compared with the LC species *S. purpurea* and *I. younghusbandii*. This finding indicates that the existing red list assessment in China primarily emphasizes population size and anthropogenic habitat pressures, while failing to account for projected threats such as climate change [14,29,37,38]. This supports the perspective shared by many researchers that conservation assessments must explicitly incorporate future climate change risks into red list evaluation frameworks [15,29,30,40,42,43].

Moreover, our analyses of PA coverage relative to species’ habitat distributions revealed generally low levels of protection across the six species in the Tibetan-Himalayan region, with none exceeding 25%. Alarmingly, protection in the most climate-vulnerable areas fell below 7%, indicating a serious conservation gap in these high-risk zones. Specifically, although *C. microphyllus* is categorized as an NT species, it occupies a relatively small proportion of high-vulnerability habitat. In contrast, *I. younghusbandii* had a significantly higher proportion of its distribution in areas of high vulnerability than *C. microphyllus*, and the former had a slightly higher percentage of protected area coverage than the latter in the areas of the highest vulnerability. These results further emphasize that in the context of future climate change, species conservation efforts must extend beyond threat assessment and instead aim for systematic, multi-dimensional strategies, including expanding PA coverage, optimizing spatial networks, and strengthening management effectiveness [43,44,47,54,65].

Human activities also significantly intensify species vulnerability, especially in densely populated cities such as Lanzhou, Chengdu, and Lhasa, where high human footprint levels lead to increased ecological pressure. This highlights the urgent need for integrated measures in such areas, such as increasing PA coverage, optimizing the spatial structure of PA networks, and enhancing management effectiveness, to establish more systematic and sustainable conservation systems [43,44,47,54,65].

From a methodological perspective, this study employed an improved CNFA to assess the vulnerability of six herbaceous species in the Tibetan-Himalayan region under future climate change. A key advancement of our approach lies in the normalization of species’ sensitivity and exposure values, which eliminates scale-related disparities and enables direct, meaningful comparisons of climate change vulnerability, both among species and across emission scenarios. Previous studies have suggested that sensitivity plays a dominant role in determining vulnerability [15,43], largely due to the use of unstandardized data. However, such conclusions may be misleading, as they tend to underestimate the critical role of exposure. By applying a normalization process, our approach ensures a balanced contribution of both sensitivity and exposure, thereby supporting a more comprehensive and integrative assessment of species vulnerability. This normalization not only standardizes feature ranges and reduces magnitude-related bias but also enhances the training efficiency and predictive accuracy of the CNFA model.

To further address the uncertainties inherent in climate model projections, we incorporated the ensemble mean of climate outputs from eight GCMs. Moreover, we evaluated the potential impacts under two representative carbon emission pathways: SSP2-4.5 (a moderate emissions scenario) and SSP5-8.5 (a high emissions scenario). Our results demonstrated that both climatic exposure and vulnerability were markedly higher under the SSP5-8.5 scenario, indicating that high carbon emissions will significantly exacerbate threats to species diversity. These findings underscore the urgent need for enhanced energy conservation and emissions reductions, not only to mitigate global climate change but also to serve as a vital strategy for biodiversity conservation [66,67].

## 4. Materials and Methods

### 4.1. Occurrence Data and Species Ranges

In this study, we focused on six plants endemic to the Tibetan-Himalayan region (73–110° E, 25–45° N): *S. kawaguchii*, *A. erecta*, *C. microphyllus*, *S. purpurea*, *I. younghusbandii*, and *A. xylorhiza*. These species are highly specialized, growing only in alpine soils above 3000 m in elevation and having an extremely narrow geographic distribution. Among them, *S. kawaguchii* belongs to the genus *Syncalathium* in the family Asteraceae, *A. erecta* belongs to the genus *Androsace* in the family Primulaceae, *C. microphyllus* belongs to the genus *Cyananthus* in the family Campanulaceae, *S. purpurea* belongs to the genus *Stipa* in the family Poaceae, *I. younghusbandii* belongs to the genus *Incarvillea* in the family Bignoniaceae, and *A. xylorhiza* belongs to the genus *Anaphalis* in the family Asteraceae. As endemic species with a restricted distribution in this region, these species belong to different families and cover different functional groups, exhibiting extreme sensitivity to climate change. Their habitat specificity makes them ideal model organisms for assessing the impact of climate change on alpine endemic species. Among these, *S. kawaguchii*, *A. erecta*, and *C. microphyllus* have been evaluated as near threatened (NT) by the Red List of Chinese Plants database (https://www.iplant.cn/rep/protlist, accessed on 2 August 2025), while *S. purpurea*, *I. younghusbandii*, and *A. xylorhiza* have been evaluated as least concern (LC). We collected occurrence records from multiple sources, including the Chinese Virtual Herbarium (CVH: https://www.cvh.ac.cn/), the Global Biodiversity Information Facility (GBIF: http://www.gbif.org/) (GBIF.org, GBIF Occurrence Download. Available online: https://doi.org/10.15468/dl.wq8pkm (accessed on 2 August 2025)), and previously published studies [68,69,70]. We then filtered the data to exclude records with incomplete information, duplicates, and those located in urban centers, retaining only records dated after 1950. To mitigate overfitting caused by spatial autocorrelation, we employed the k-nearest neighbors technique [71], ensuring that only one sample point was retained within a 5 km radius. The final samples size for modeling ranged from 21 to 102 occurrences per species (Appendix A), with their spatial distribution shown in Figure 7.

Then, we estimated the distribution ranges of six plant species using inverse distance weighting (IDW), which is a geospatial model that relies only on species occurrence sample points and can be implemented using the “geoIDW” function in R [72]. The distribution ranges of these species approximate their respective habitats for subsequent vulnerability assessments [73].

### 4.2. Bioclimatic Variables

First, we downloaded 19 current (representing averages from 1970 to 2000) and future (2050, representing the average from 2040 to 2060) bioclimatic variables from the WorldClim v2.1 database (https://www.worldclim.org/) at a spatial resolution of 2.5 arc-minutes. For the future climate, due to climate uncertainty, we used an integration approach of eight global climate models (GCMs). Specifically, we used the following eight GCMs: BCC-CSM2-MR, CNRM-CM6-1, CNRM-ESM2-1, CanESM5, IPSL-CM6A-LR, MIROC-ES2L, MIROC6, and MRI-ESM2-0. We then computed their averages as the climate data for 2050. In addition, the future climate data are based on two shared socioeconomic pathways (SSPs) provided by the Intergovernmental Panel on Climate Change (IPCC) Sixth Assessment Report (AR6), namely (1) SSP2-4.5, which represents moderate CO_2_ emissions and a global temperature rise of 3.8–4.2 °C by the end of the 21st century, and (2) SSP5-8.5, which represents the high CO_2_ emission scenario, with a projected global temperature increase of 4.7–5.1 °C by the end of the 21st century [74]. Subsequently, the “crop” and “mask” functions in R were used to crop the bioclimatic variables to the study area, followed by correlation analysis. Bioclimatic variables with a Pearson correlation greater than 0.7 were excluded, and the final set of bioclimatic variables selected for modeling included the mean diurnal range (MDR), isothermality (ISO), mean temperature of the warmest quarter (MTWQ), precipitation seasonality (PS), precipitation of the warmest quarter (PWQ), and precipitation of the coldest quarter (PCQ).

### 4.3. Protected Areas and Human Footprint Data

In this study, we also considered the impacts of protected areas and human footprints on species vulnerability in the context of future climate change. We integrated two parts of publicly available protected area data: one from Protected Planet on discovering the world’s protected and conserved areas (https://www.protectedplanet.net/) and the other from the China Nature Reserve Biospecimen Resource Sharing Platform (China Nature Reserve Specimen Resource Sharing Platform. (2024). List and Vector Boundaries of Nature Reserves in China. Zenodo. https://doi.org/10.5281/zenodo.14875797 (accessed on 2 August 2025)). The relevant protected area data for the study area were then extracted [75]. Meanwhile, the human footprint data, obtained from the Socio-economic Data and Application Center (SEDAC: https://www.earthdata.nasa.gov/centers/sedac-daac (accessed on 2 August 2025)), are commonly used in natural resource management and conservation research. They are based on population density, land use changes, and infrastructure data collected between 1960 and 2001, providing an estimate of the human impact on the natural environment and illustrating the degree of human interference with the Earth’s ecosystems. The map shows that human footprint values are significantly higher in major densely populated urban areas around the world, such as the Tibetan-Himalayan region megacities (including Lhasa, Chengdu, Lanzhou, and New Delhi, among others).

### 4.4. Climate Change Vulnerability Assessments

In this study, the vulnerability of six typical plants in the Tibetan-Himalayan region to future climate change was assessed using improved Climate Niche Factor Analysis (CNFA) [40]. Based on the assumption of ecological niche conservatism, this method analyzes the intrinsic sensitivity and external exposure of species to climate change under future climate scenarios and normalizes the above two indicators to conduct a comprehensive assessment of species vulnerability.

#### 4.4.1. Sensitivity

Sensitivity is defined as the degree to which a species’ ability to survive is constrained by the climatic conditions of its habitat. Typically, the more a species is constrained by the climatic conditions within its current range, the more sensitive it is to future climate change. Accordingly, the species’ climate niche tends to be smaller [40]. To quantify the ecological niche of each species, we compared the distribution of species in ecological space with the global distribution of available environmental conditions [40,76]. Following Rinnan and Lawler (2019) [40], we quantified two aspects of species ecological niches: (1) marginality, which reflects the distance between the ecological niche center of gravity of the climatic conditions of the species’ habitat and the entire study area; and (2) specialization, which indicates the ratio between the global ecological niche and the size of the species’ ecological niche. To define the global distribution, we used the combined range of the six plant species in the Tibetan-Himalayan region as the N cells; that is, the extent of the combined range was from 73° E to 110° E and from 25° N to 40° N. For the distribution of each Tibetan-Himalayan region plant species with N cells, we used both estimated range and species occurrence records generated by the IDW method to represent the distribution of each species. At the same time, we performed the CNFA test for both methods to verify the robustness of the results. Then, in the P-dimensional ecological space, the marginality factor *m_i_* and the specialization factor $\left(u_{i1},u_{i2},\cdots ,u_{ip-1}\right)$ were first combined as $\left(m_{i},u_{i1},\cdots ,u_{ip-1}\right)$, and then normalized as $\left(w_{i1},w_{i2},\cdots ,w_{ip}\right)$. After that, the sensitivity factor $s_{i}$ corresponding to each bioclimatic variable *i* was calculated by the equation $s_{i}=\sum_{k=1}^{P}w_{ik}\rho_{k}$, where $\rho_{1}$ refers to the amount of specialization on the liminality axis and $\rho_{k}\left(k>1\right)$ is the amount of specialization on the specialization axis. Next, the total sensitivity of each species was calculated using the equation $S=\sqrt{1/p\sum_{i=1}^{p}s_{i}}$. The higher the climate sensitivity of a species, the higher its vulnerability to climate change [40]. We used the “cnfa” function in the CENFA package to calculate the marginality, specialization, and sensitivity.

#### 4.4.2. Exposure

Exposure is defined as the degree of climate change experienced by a species within its current range. To calculate indicators of species’ exposure to climate change, we used a measure of dissimilarity between current and future climate conditions within the species’ range. Specifically, the exposure factor under future climate change was first calculated as $d_{j}=\sum_{i=1}^{N}p_{i}\left|g_{ij}-z_{ij}\right|$, where $g_{ij}$ denotes the value of the future bioclimatic variable *j* at location *i*, $z_{ij}$ denotes the value of the current bioclimatic variable *j* at location *i*, and $p_{i}$ is the habitat utilization at location *i*. The overall indicator of exposure is $D=\sqrt{\sum_{j=1}^{p}d_{j}^{2}}$. The higher the climate exposure of a species, the greater the deviation of its habitat between the current and future climates [40]. We calculated the exposure of each species under future climate scenarios using the “departure” function in the CENFA package.

#### 4.4.3. Vulnerability

Without considering the potential impacts of adaptive capacity, climate change vulnerability reflects the interaction between species sensitivity and exposure [40]. In order to facilitate the comparison of vulnerability among the six species and the same species under different carbon emission pathways, we carried out a max-min normalization process on the sensitivity factors of all species and the exposure factors under different carbon emission pathways. This was realized by the following equation: $s_{j}^{\prime }=s_{j}-s_{j\min}/s_{j\max}-s_{j\min}$, $d_{j}^{\prime }=d_{j}-d_{j\min}/d_{j\max}-d_{j\min}$. Here, $s_{j}^{\prime }\;$ and $d_{j}^{\prime }\;$ represent the normalized sensitivity and exposure factors, respectively, $s_{j\max}$ and $s_{j\min}$ represent the maximum and minimum values of the sensitivity factor for all species, respectively, and $d_{j\max}$ and $d_{j\min}$ represent the maximum and minimum values of the exposure factor for all species, respectively. Thus, the vulnerability factor is $v_{j}=\sqrt{\left(1+d_{j}^{\prime }\right)s_{j}^{\prime }}$, and the overall indicators of vulnerability are $V=\sqrt{1/p\sum_{j=1}^{p}v_{j}}$. We calculated each species’ vulnerability to climate change as a geometric mean of the normalized sensitivity and exposure using the “vulnerability” function in the CENFA package. Similar to exposure, we calculated the vulnerability of each species under different future climate scenarios. Through the above approach, the sensitivity factor and exposure factor indicators for all species were scaled to a range from 0 to 1, eliminating their differences in magnitude and ensuring that sensitivity and exposure contributed equally to the species vulnerability analysis [77]. Finally, the vulnerability factors and overall indicators of vulnerability of species were calculated according to the above formulas, and the differences in vulnerability between different species and the same species under different carbon emission pathways were further compared.

### 4.5. Spatial Vulnerability Analysis

Based on the filtered species distribution data and climate variable data, we used the “predict” function in the CENFA package to project the sensitivity, exposure, and vulnerability metrics of the species onto their respective habitats, thus obtaining the spatial sensitivity, exposure, and vulnerability of each species. In addition, we mapped the spatial sensitivity, exposure, and vulnerability of each species using the “sensitivity_map”, “exposure_map”, and “vulnerability_map” functions to map the spatial distribution patterns of the sensitivity, exposure, and vulnerability of each species [40]. To study the spatial vulnerability of the six plant species, we calculated the sensitivity, exposure, and vulnerability of each grid cell. Eventually, we normalized the sensitivity, exposure, and vulnerability indicators for all grid cells to a range from 0 to 1, forming a system of spatial sensitivity, exposure, and vulnerability indicators. Among them, spatial vulnerability was classified into three levels: low (V: 0–1/3), medium (V: 1/3–2/3), and high (V: 2/3–1) [42].

### 4.6. Impacts of Protected Areas and Human Footprints on Species Vulnerability

First, we extracted masks based on the spatial surface data of protected areas with species vulnerability distribution maps and obtained the coverage rates of protected areas for each species by calculating the ratio of the protected area to the total area of the habitat within the species’ habitat range. Subsequently, based on the three grades of low, medium, and high vulnerability, we counted the proportion of protected area under each grade and then assessed the impact of protected area distribution on species vulnerability. Next, we resampled the human footprint data to the same resolution as the species spatial vulnerability and normalized each grid cell in the human footprint raster data to a value range between 0 and 1 (Appendix A). Finally, we analyzed the human footprint data with spatial vulnerability distributions of species in a mask overlay and extracted species habitat-wide validated data to assess the impact of human footprints on species vulnerability. The above masking, resampling, and overlaying operations are all available in ArcGIS (version 10.8) and RStudio (version 4.3.1).

## 5. Conclusions

In this study, we applied an improved CNFA framework to assess the vulnerability of six alpine endemic herbaceous plants in the Tibetan-Himalayan region to climate change and human activities. The key findings are as follows:The species patterns of climate change vulnerability revealed a high in the west and low in the east distribution, with the central Tibetan-Himalayan region and HDMs potentially acting as important future climate refugia.Species exposure and vulnerability were significantly higher under high carbon emission scenarios (SSP5-8.5) compared with moderate carbon emission scenarios (SSP2-4.5), emphasizing the critical role of climate policy in mitigating biodiversity loss.The existing PA coverage is alarmingly inadequate, with less than 25% of our studied species’ habitats currently under protection and coverage dropping below 7% in highly vulnerable zones. Species vulnerability is further exacerbated in areas with dense human activity, such as the areas surrounding Lhasa and Chengdu.Limitations in current conservation assessments were observed, as the IUCN Red List does not sufficiently account for future climate risks. For example, *S. purpurea* (classified as LC) may be more vulnerable to climate change than *C. microphylla* (classified as NT), highlighting discrepancies in current threat classifications.

Overall, this study underscores the urgency of integrating climate change projections into global and national conservation assessment frameworks. By quantifying species vulnerability, identifying conservation gaps, and evaluating human pressures, our findings provide a scientific basis for developing targeted, climate-adaptive conservation strategies for alpine endemic plant species in the Tibetan-Himalayan region.

## Figures and Tables

**Figure 1 plants-14-02424-f001:**
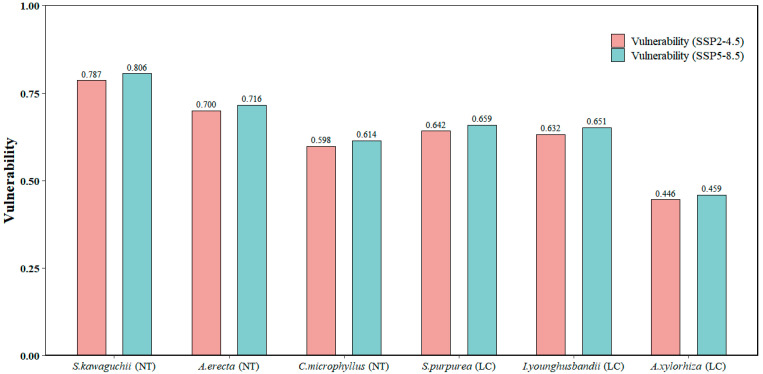
Vulnerability of six plant species to two different carbon emission pathways (SSP2-4.5 and SSP5-8.5) on the Tibetan-Himalayan region after normalization. The symbols in parentheses correspond to the endangered status of each species.

**Figure 2 plants-14-02424-f002:**
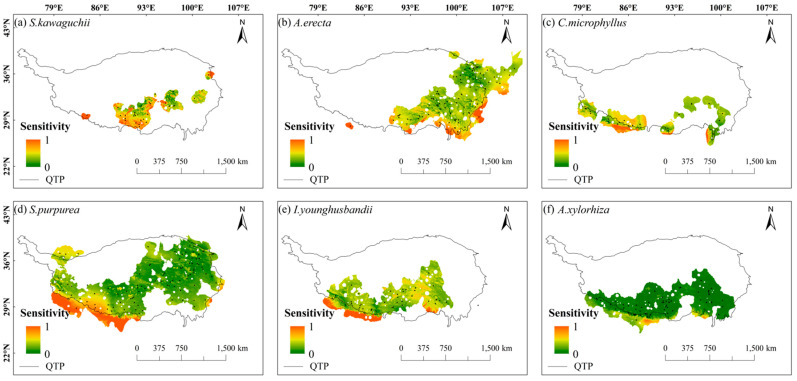
Sensitivity of six plant species in the Tibetan-Himalayan region. The black dots in the figure represent the distribution of sample recording sites for the six plant species.

**Figure 3 plants-14-02424-f003:**
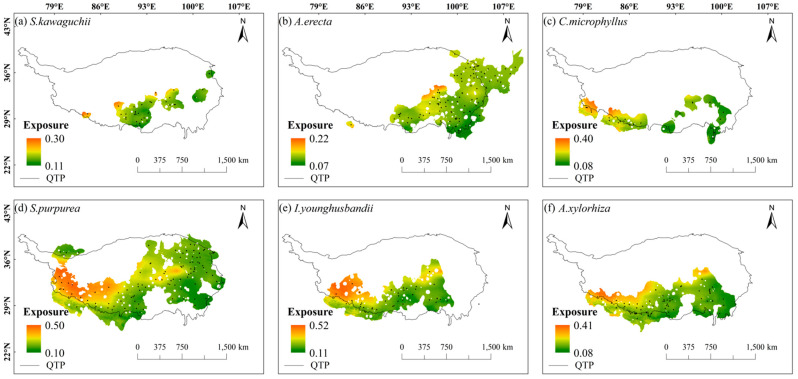
Exposure of six plant species in the Tibetan-Himalayan region under future climate change scenarios (shared socioeconomic pathways; SSP2-4.5). The black dots in the figure represent the distribution of sample recording sites for the six plant species.

**Figure 4 plants-14-02424-f004:**
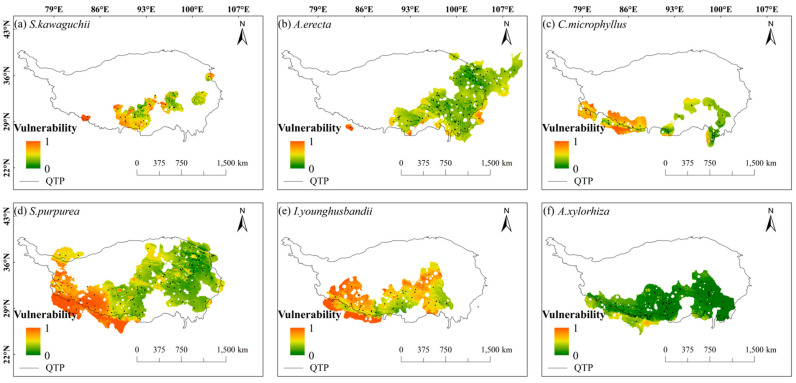
Vulnerability of six plant species in the Tibetan-Himalayan region under future climate change scenarios (shared socioeconomic pathways; SSP2-4.5). The black dots in the figure represent the distribution of sample recording sites for the six plant species.

**Figure 5 plants-14-02424-f005:**
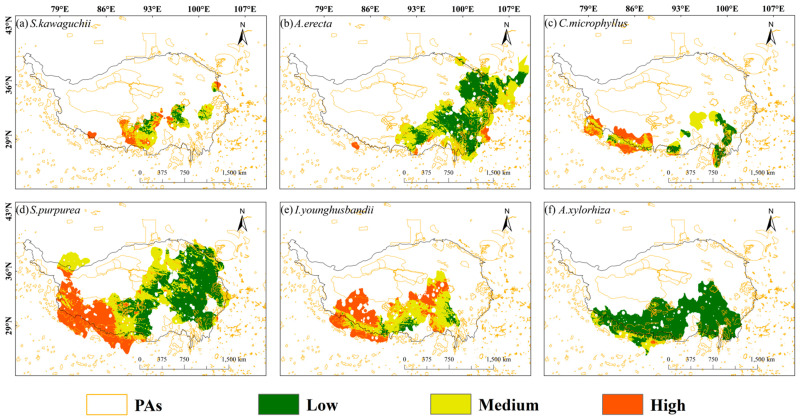
Under a future climate change scenario (shared socioeconomic pathways; SSP2-45), the vulnerability of six plant species in the Tibetan-Himalayan region was classified as low, medium, or high and visualized through a color gradient: dark green for low vulnerability, light green for medium vulnerability, and red for high vulnerability. In addition, the areas marked with yellow outlines in the figure indicate the extent of protected areas, which were used to analyze the impact of protected areas on species vulnerability.

**Figure 6 plants-14-02424-f006:**
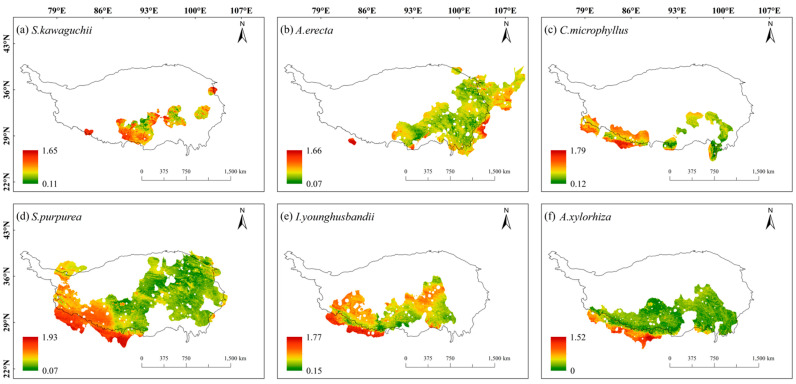
Vulnerability of six plant species in the Tibetan-Himalayan region under future climate change scenarios (shared socioeconomic pathways; SSP2-4.5) coupled with human footprints.

**Figure 7 plants-14-02424-f007:**
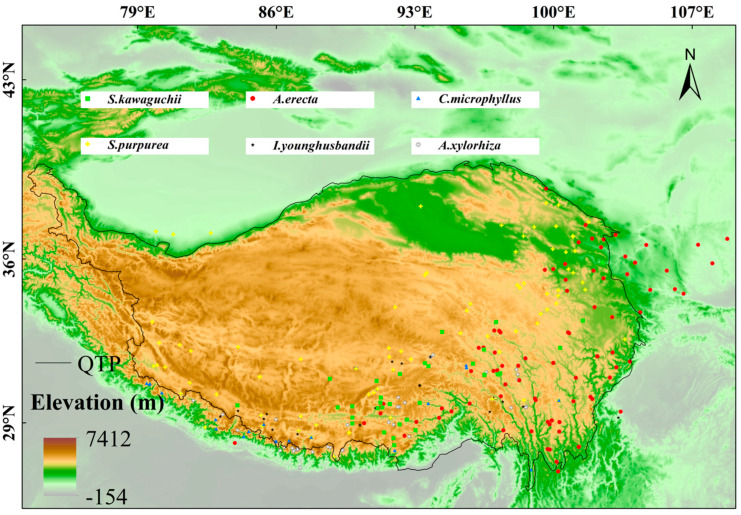
Occurrence data and distribution of six plant species in the Tibetan-Himalayan region.

**Table 1 plants-14-02424-t001:** Sensitivity factors for six plant species, as well as exposure and vulnerability factors under different shared socioeconomic pathway (SSP) scenarios. Coefficients marked in bold in each column indicate the two largest values corresponding to each species.

Species	Bioclimate	Sensitivity	Exposure		Vulnerability	
			SSP2-4.5	SSP5-8.5	SSP2-4.5	SSP5-8.5
*S. kawaguchii*	MDR	**13.58**	**0.22**	**0.31**	**4.07**	**4.22**
	ISO	11.19	0.14	0.17	3.57	3.62
	MTWQ	6.08	**0.30**	**0.44**	2.81	2.96
	PS	4.44	0.05	0.08	2.16	2.19
	PWQ	**30.62**	0.09	0.16	**5.78**	**5.96**
	PCQ	7.13	0.01	0.01	2.68	2.69
*A. erecta*	MDR	4.35	**0.13**	**0.19**	2.22	2.28
	ISO	3.06	0.09	0.14	1.83	1.87
	MTWQ	**11.89**	**0.30**	**0.44**	**3.93**	**4.14**
	PS	8.88	0.06	0.09	3.07	3.11
	PWQ	**14.50**	0.07	0.10	**3.93**	**3.99**
	PCQ	7.23	0.02	0.03	2.71	2.72
*C. microphyllus*	MDR	4.13	**0.20**	**0.28**	2.23	2.30
	ISO	3.50	0.19	0.27	2.04	2.11
	MTWQ	4.24	**0.28**	**0.42**	2.33	2.45
	PS	**6.54**	0.14	0.21	**2.74**	**2.81**
	PWQ	**9.22**	0.13	0.19	**3.23**	**3.32**
	PCQ	2.66	0.04	0.04	1.67	1.66
*S. purpurea*	MDR	5.41	0.23	**0.33**	2.58	2.68
	ISO	4.82	**0.24**	0.33	2.44	2.53
	MTWQ	5.15	**0.33**	**0.48**	**2.62**	**2.76**
	PS	3.97	0.12	0.17	2.11	2.16
	PWQ	**8.70**	0.09	0.14	**3.09**	**3.15**
	PCQ	**6.55**	0.01	0.02	2.58	2.58
*I. younghusbandii*	MDR	3.17	**0.26**	**0.37**	2.00	2.09
	ISO	5.32	0.26	0.34	2.59	2.67
	MTWQ	**6.15**	**0.32**	**0.47**	**2.85**	**3.00**
	PS	4.94	0.14	0.21	2.37	2.45
	PWQ	**14.75**	0.12	0.18	**4.06**	**4.17**
	PCQ	2.30	0.02	0.02	1.53	1.53
*A. xylorhiza*	MDR	3.30	**0.21**	**0.30**	2.00	2.07
	ISO	2.74	0.19	0.25	1.81	1.85
	MTWQ	**3.78**	**0.29**	**0.43**	**2.21**	**2.32**
	PS	2.28	0.10	0.15	1.58	1.62
	PWQ	**3.91**	0.12	0.18	**2.09**	**2.15**
	PCQ	1.93	0.02	0.03	1.41	1.41

**Table 2 plants-14-02424-t002:** Total indicators of marginality and sensitivity for the six plant species, as well as total indicators of exposure and vulnerability under different shared socioeconomic pathway (SSP) scenarios. The total indicators of vulnerability are normalized values.

Species	Marginality	Sensitivity	Exposure		Vulnerability	
			SSP2-4.5	SSP5-8.5	SSP2-4.5	SSP5-8.5
*S. kawaguchii*	1.241	3.489	0.409	0.593	0.787	0.806
*A. erecta*	0.667	2.884	0.350	0.520	0.700	0.716
*C. microphyllus*	3.148	2.247	0.441	0.638	0.598	0.614
*S. purpurea*	1.033	2.402	0.492	0.707	0.642	0.659
*I. younghusbandii*	1.810	2.470	0.523	0.741	0.632	0.651
*A. xylorhiza*	1.919	1.730	0.437	0.626	0.446	0.459

**Table 3 plants-14-02424-t003:** Area share of six plant species in different shared socioeconomic pathway (SSP) scenarios and in three vulnerability classes. Values in parentheses indicate the area share of protected areas in each vulnerability class.

Area (%)		Low	Medium	High	PAs
*S. kawaguchii*	SSP2-4.5	17.6 (0.89)	54.3 (4.41)	28.1 (3.25)	8.55
	SSP5-8.5	14.8 (0.81)	56.9 (4.54)	28.3 (3.20)	8.55
*A. erecta*	SSP2-4.5	48.7 (4.70)	48.2 (7.11)	3.1 (0.67)	12.48
	SSP5-8.5	38.6 (3.57)	57.8 (8.16)	3.6 (0.75)	12.48
*C. microphyllus*	SSP2-4.5	26.6 (8.59)	43.5 (10.71)	29.9 (5.72)	25.02
	SSP5-8.5	22.1 (8.11)	48.1 (11.26)	29.8 (5.65)	25.02
*S. purpurea*	SSP2-4.5	40.9 (4.42)	33.8 (6.08)	25.3 (5.02)	15.52
	SSP5-8.5	38.7 (4.28)	36.1 (6.11)	25.2 (5.13)	15.52
*I. younghusbandii*	SSP2-4.5	7.5 (1.10)	48.9 (8.24)	43.6 (6.95)	16.29
	SSP5-8.5	8.3 (0.90)	48.9 (8.82)	42.8 (6.57)	16.29
*A. xylorhiza*	SSP2-4.5	91.5 (17.80)	8.1 (1.42)	0.4 (0.01)	19.23
	SSP5-8.5	91.5 (17.75)	8.4 (1.48)	0.1 (0.00)	19.23

## Data Availability

Data will be made available on request.

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
