# Peer review of "Vulnerability Assessment of Six Endemic Tibetan-Himalayan Plants Under Climate Change and Human Activities"

_plants, 2025, doi:10.3390/plants14152424_

Round 1
Reviewer 1 Report
Comments and Suggestions for Authors
Dear all. After reading the manuscript entitled "Vulnerability Assessment of Six Endemic Tibetan-Himalayan Plants under Climate Change and Human Activities," After carefully reading the manuscript, I had a favorable impression of it. Below, I will include some specific comments.
The manuscript focuses on the interplay between climate change and anthropogenic pressures. By integrating climate niche modeling with human footprint data, the research aims to discern vulnerability patterns that can inform targeted conservation strategies. The Tibetan-Himalayan region, recognized as a global biodiversity hotspot, is currently experiencing rapid environmental transformation, making this study a significant contribution to understanding the local impacts of climate change on endemic species.
The application of an enhanced Climate Niche Factor Analysis (CNFA) that incorporates both climate sensitivity and human footprint metrics is particularly newsworthy. The integration of normalization techniques with spatial overlays and anthropogenic impact data addresses a significant methodological gap that has been overlooked in the literature.
The methodology is fundamentally sound and builds on a robust theoretical framework. However, the manuscript would benefit from elaborating on the selection criteria for the six species under study. Are these species phylogenetically diverse or representative of specific functional groups, or is their inclusion simply a consequence of data availability? While this information is not strictly essential, it would enhance the manuscript’s clarity.
The tables and figures presented are well-chosen and effectively contribute to the overall narrative. The conclusions drawn align well with the evidence provided, particularly regarding the spatial vulnerability gradients identified and the discrepancies between species vulnerability and protected area (PA) coverage. The assertion that IUCN Red List assessments “do not adequately account for future climate risks” is a valid concern in general; however, this claim would benefit from comparative analysis. For instance, it would be helpful to indicate whether the species evaluated are currently listed on the IUCN Red List and if their existing statuses accurately reflect the projected future risks discussed in this study.
Lastly, while the references cited are generally suitable, the manuscript would be strengthened by a more comprehensive engagement with relevant literature in these areas better to situate the findings within the broader research context.
Author Response
- Response to comment:(Are these species phylogenetically diverse or representative of specific functional groups, or is their inclusion simply a consequence of data availability?)
Response: We reviewed relevant information on these six herbaceous plants and found that they belong to different functional groups. We selected these six species because they are endemic to the Tibetan-Himalayan, are highly susceptible to climate change, and their distribution points are readily accessible. Some of these species are NT, while others are LC. By using the improved CNFA to assess the vulnerability of these plants to climate change, we have further emphasized the necessity of incorporating climate change predictions into the IUCN Red List conservation criteria. A brief explanation of this approach is provided in the Materials and Methods section (lines 362–372).
- Response to comment:(The assertion that IUCN Red List assessments “do not adequately account for future climate risks” is a valid concern in general; however, this claim would benefit from comparative analysis.)
Response: Yes, the assertion that the IUCN Red List assessments “do not adequately account for future climate risks” benefits from comparative analysis. We presented the endangered status of these six species in the Materials and Methods section (lines 372–375) and described the differences in their vulnerability after considering future climate change in the Abstract (lines 21–23) and Results section (lines 149–155). Additionally, we have labeled the endangered status of each species next to their names in Figure 1, making it clear that the LC species S. purpurea exhibits higher vulnerability than the NT species C. microphyllus when climate change is taken into account.
- Response to comment:(Lastly, while the references cited are generally suitable, the manuscript would be strengthened by a more comprehensive engagement with relevant literature in these areas better to situate the findings within the broader research context.)
Response: We supplemented our review of the relevant literature and added new references to the Introduction and Discussion sections to make the literature review of the relevant research field more comprehensive. Newly added literature is highlighted in yellow in the References.
Reviewer 2 Report
Comments and Suggestions for Authors
The manuscripts brings interesting and novel views on species survival under various climate scenarios and implication in target species conservation.
The main concern of the manuscript is that the authors did not taken into consideration species biology at present, i.e. if the species are living in suboptimal or sublethal conditions and are they spreading vegetative or the sexually, if some of the developmental stage are missing or not.
Technically, the application of the prediction models seems to be OK. There are inconsistence in using the terms defined by IUCN like vulnerable/ endangered/ threaten (with derivatives like the most vulnerable, highly endangered etc...) and it is rather hard to follow the meaning since they applied are throughout the text. It needs to be refine. They also applied highly endangered which is critically endangered in IUCN. This needs to be clear for all the readers.
The IUCN criteria are mentioned in many places colloquaily. For example, it should be assesmen on IUCN Red Listing Critheria (not Red List) in line 65. Please check and be precise in all manuscript.
L. 69-71. not focus but primarily missing the projection on populations, modeling... That is why criterion C is not used often.
Species names should be written fully where they appear firstly within the text with authorities names as well. The italic should be applied for latin names. It is no so in most of the manuscript.
Subtitles should start with capital letters.
Finally, the title change is suggested to:
Threat assessment...
Comments on the Quality of English LanguageThe English is generally fine, but in many places it needs small improvements. E.g. most vulnerable to the most vulnerable etc...
Author Response
- Response to comment:(The main concern of the manuscript is that the authors did not take into consideration species biology at present)
Response: The purpose of this article is to apply the improved CNFA to characterize the vulnerability of these six species in the Tibetan-Himalayan under future climate change and human activities. We provide a brief description of these six species in the Materials and Methods section (lines 362–372).
- Response to comment:(There are inconsistence in using the terms defined by IUCN like vulnerable/ endangered/ threaten (with derivatives like the most vulnerable, highly endangered etc...) and it is rather hard to follow the meaning since they applied are throughout the text.)
Response: We have made improvements to areas where there were inconsistencies in the use of terms defined by the International Union for Conservation of Nature (IUCN) throughout the text. For example, we have replaced “high vulnerability” with “significant vulnerability” to avoid confusion with terms defined by the IUCN. And highlight all replaced terms in yellow.
- Response to comment:(The IUCN criteria are mentioned in many places colloquially. For example, it should be assessment on IUCN Red Listing Criteria (not Red List) in line 65. Please check and be precise in all manuscript.)
Response: We have replaced the IUCN Red List in line 65 with the IUCN Red List Criteria, and the sentence has been moved to line 70. We have also revised the language throughout the text.
- Response to comment:( 69-71. not focus but primarily missing the projection on populations, modeling... That is why criterion C is not used often.)
Response: The current IUCN Red List assessment system has a key flaw, namely insufficient consideration of the impact of climate change, which is precisely the core issue addressed in this study. Although the IUCN criterion C (for small population size and decline) you mentioned is indeed rarely used, reflecting the limitations of the system in terms of its focus on species population dynamics, the focus of this paper is to reveal the shortcomings of the assessment system in terms of its consideration of climate change factors.
- Response to comment:(Species names should be written fully where they appear firstly within the text with authorities’ names as well. The italic should be applied for Latin names. It is no so in most of the manuscript.)
Response: We have written out the species name in full where it first appears in the article and indicated the name of the responsible authority (lines 121-126). We have also changed the Latin names of species throughout the text to italics.
- Response to comment:(Subtitles should start with capital letters.)
Response: We have capitalized the 411 lines of subtitles.
- Response to comment:(Finally, the title change is suggested to:Threat assessment...)
Response: This article mainly discusses the vulnerability of species to climate change and human activities. The title “Threat” is too broad. The vulnerability of species to climate change refers to their intrinsic sensitivity and extrinsic exposure to climate change. This article also calculates the sensitivity and exposure indices of various species. In addition, the related literature below also calculates the vulnerability of species, and their titles are all written as “vulnerability” .
- Jamwal, P.S.; Di Febbraro, M.; Carranza, M.L.; Savage, M.; Loy, A. Global Change on the Roof of the World: Vulnerability of Himalayan Otter Species to Land Use and Climate Alterations. Divers. Distrib. 2022, 28, 1635–1649, doi:10.1111/ddi.13377.
- Wang, W.-T.; Guo, W.-Y.; Jarvie, S.; Serra-Diaz, J.M.; Svenning, J.-C. Anthropogenic Climate Change Increases Vulnerability of Magnolia Species More in Asia than in the Americas. Biol. Conserv. 2022, 265, 109425, doi:10.1016/j.biocon.2021.109425.
Reviewer 3 Report
Comments and Suggestions for Authors
Estimating the future distribution of species in response to climate change is of general population ecology and general ecological importance. Since the selected species are of particular importance for nature conservation, knowledge of future trends is all the more important for active nature conservation management.
The methodological approach is consistent with current practice.
The niche models are based on current and future climate parameters, information on the protected status of the areas, and anthropogenic footprint data. Thus, the approach goes beyond classic climate niche models.
However, the following notes are appropriate:
1. The species studied should be briefly characterized in the text, especially their key biological and ecological characteristics. This should also include information on their habitat requirements! Important characteristics include life form, vegetative and/or generative dispersal capacity, and soil requirements (pH spectrum).
2. This last point leads to an important set of data that was not considered in the models: soil conditions. Which soil types are the species dependent on? This information would further refine the niche models.
Author Response
1. Response to comment: (The species studied should be briefly characterized in the text, especially their key biological and ecological characteristics. This should also include information on their habitat requirements! Important characteristics include life form, vegetative and/or generative dispersal capacity, and soil requirements (pH spectrum))
Response: We reviewed relevant information on these six herbal plants and found that they belong to different functional groups. We have provided a brief description in the Materials and Methods section (lines 362-372).
2. Response to comment:(This last point leads to an important set of data that was not considered in the models: soil conditions. Which soil types are the species dependent on? This information would further refine the niche models.)
Response: Thank you for your valuable comment. Our primary focus in this study was on the impacts of climate change and human activities on species vulnerability, which is why soil conditions were not considered in our models. Additionally, to the best of our knowledge, there is currently no reliable future soil data available. If we were to include soil conditions, it would require assuming that future soil data will remain the same as current data, which may lead to an underestimation of the impact of soil changes on species vulnerability. This assumption could result in unreliable conclusions, so we have opted not to consider soil factors in this analysis.
Reviewer 4 Report
Comments and Suggestions for Authors Dear Authors, After reading the manuscript, I have several suggestions and comments that should improve the quality of the content presented. These are: 1. The abstract lacks a clear research objective and a hypothesis or research question. Therefore, the abstract should be supplemented with these elements. 2. The "Introduction" chapter should also be expanded and thematically divided to clarify why the authors undertook this particular study and what research gap they are filling in this regard. Furthermore, a research question and/or hypothesis should be included at the end of this chapter. 3. The "Introduction" chapter should be followed by a "Research Methods" chapter, followed only by a "Research Results" chapter. Please reverse the order of these chapters. 4. Both the "Research Methodology" and "Research Results" chapters are well-described and organized thematically into subsections. 5. Therefore, it is even more desirable to organize and expand the "Introduction," as well as organize and expand the "Discussion" chapter. This will make the content coherent and clearer. 6. There is no "Conclusions" chapter, which should briefly present, in bullet points, the findings from the authors' research. This chapter will be easy to write if the authors consider my suggestions for organizing the aforementioned chapters. Furthermore, the conclusions should conclude with a statement of future research directions resulting from this research. 7. Regarding the bibliography, it should be supplemented with at least 20 items, which should be included in a well-organized and well-organized "Introduction" chapter and a well-organized and well-organized "Discussion" chapter. Furthermore, the literature should be no older than 2010, unless a work older than 2010 is considered classic and fundamental for the given topic. After considering the above comments and suggestions, the article can be published. Good luck!Author Response
- Response to comment:(The abstract lacks a clear research objective and a hypothesis or research question.)
Response: We have rewritten the research objectives and research questions in the abstract section (lines 7-23).
- Response to comment:(The "Introduction" chapter should also be expanded and thematically divided to clarify why the authors undertook this particular study and what research gap they are filling in this regard. Furthermore, a research question and/or hypothesis should be included at the end of this chapter.)
Response: The objective of our study was to focus on six representative alpine herbaceous plants endemic to the Tibetan-Himalayan region and to explore the vulnerability of species in this region to the combined effects of climate change and human activities (lines 103–105). The study primarily addressed two gaps: first, it identified the impacts of human activities, defined priority conservation areas, and pointed out the shortcomings of current conservation measures (lines 110–115). Second, it emphasizes the urgency of incorporating climate change projections into global and national conservation assessment frameworks (lines 73–75).
- Response to comment:(The "Introduction" chapter should be followed by a "Research Methods" chapter, followed only by a "Research Results" chapter. Please reverse the order of these chapters.)
Response: Thank you for your comment. We originally followed the order of Introduction, Materials and Methods, Results, and Discussion. However, based on the template requirements of Plants journal, we revised the chapter sequence to place the Results and Discussion sections before the Methods chapter, as per the journal's guidelines.
- Response to comment:(4. Both the "Research Methodology" and "Research Results" chapters are well-described and organized thematically into subsections. 5. Therefore, it is even more desirable to organize and expand the "Introduction," as well as organize and expand the "Discussion" chapter.)
Response: We have expanded and refined the Introduction and Discussion sections.
- Response to comment:(There is no "Conclusions" chapter, which should briefly present, in bullet points, the findings from the authors' research. This chapter will be easy to write if the authors consider my suggestions for organizing the aforementioned chapters.)
Response: We have written the conclusion section of this article in small paragraphs (lines 521-543).
- Response to comment:(Regarding the bibliography, it should be supplemented with at least 20 items, which should be included in a well-organized and well-organized "Introduction" chapter and a well-organized and well-organized "Discussion" chapter. Furthermore, the literature should be no older than 2010, unless a work older than 2010 is considered classic and fundamental for the given topic.)
Response: We have supplemented and updated the literature, retaining some important literature from before 2010 and adding several new references. The new references are marked in yellow in the references section.
Round 2
Reviewer 3 Report
Comments and Suggestions for Authors
My remarks were well included in the new manuscript.